# Influence of Cu Content on Structure and Magnetic Properties in Fe_86-x_Cu_x_B_14_ Alloys

**DOI:** 10.3390/ma13061451

**Published:** 2020-03-23

**Authors:** Tymon Warski, Patryk Wlodarczyk, Marcin Polak, Przemyslaw Zackiewicz, Adrian Radon, Anna Wojcik, Maciej Szlezynger, Aleksandra Kolano-Burian, Lukasz Hawelek

**Affiliations:** 1Lukasiewicz Research Network—Institute of Non-Ferrous Metals, 5 Sowinskiegostr., 44-100 Gliwice, Poland; patrykw@imn.gliwice.pl (P.W.); marcin.polak@imn.gliwice.pl (M.P.); przemyslaw.zackiewicz@imn.gliwice.pl (P.Z.); adrian.radon@imn.gliwice.pl (A.R.); olak@imn.gliwice.pl (A.K.-B.); 2Institute of Metallurgy and Materials Science Polish Academy of Sciences, 25 Reymonta str., 30-059 Krakow, Poland; wojcik.a@imim.pl (A.W.); m.szlezynger@imim.pl (M.S.)

**Keywords:** soft magnetic materials, metallic glass, crystallization, magnetic properties

## Abstract

Influence of Cu content on thermodynamic parameters (configurational entropy, Gibbs free energy of mixing, Gibbs free energy of amorphous phase formation), crystallization kinetics, structure and magnetic properties of Fe_86-x_Cu_x_B_14_ (x = 0, 0.4, 0.55, 0.7, 1) alloys is investigated. The chemical composition has been optimized using a thermodynamic approach to obtain a minimum of Gibbs free energy of amorphous phase formation (minimum at 0.55 at.% of Cu). By using differential scanning calorimetry method the crystallization kinetics of amorphous melt-spun ribbons was analyzed. It was found that the average activation energy of α-Fe phase crystallization is in the range from 201.8 to 228.74 kJ/mol for studied samples. In order to obtain the lowest power core loss values, the isothermal annealing process was optimized in the temperature range from 260 °C to 400 °C. Materials annealed at optimal temperature had power core losses at 1 T/50 Hz—0.13–0.25 W/kg, magnetic saturation—1.47–1.6 T and coercivity—9.71–13.1 A/m. These samples were characterized by the amorphous structure with small amount of α-Fe nanocrystallites. The studies of complex permeability allowed to determine a minimum of both permeability values at 0.55 at.% of Cu. At the end of this work a correlation between thermodynamic parameters and kinetics, structure and magnetic properties were described.

## 1. Introduction

Fe-based alloys, such as METGLAS, FINEMET, NANOPERM or HITPERM are widely used as a replacement for silicon steel in electricity generation and in magnetic, electronic and conversion applications. This is associated with wider usable frequency range, lower coercivity, weight and power losses [1]. For many years the chemical composition of soft magnetic alloys has been modified by various combinations of atomic substitutions, however, the development of new production techniques, especially postprocessing treatment (like ultra-rapid annealing—URA), has increased interest in binary alloys like Fe-B. This binary alloy is known for its superior glass forming ability (GFA) [2]. The high value of GFA parameter is necessary to prepare amorphous alloys by rapid quenching and its further processing. Content of B has a crucial role in that process due to negative mixing enthalpy of Fe-B binary alloy equal to −35.5 kJ/mol [3,4]. The main disadvantage of higher B content is a tendency to formation borides which results in increasing power losses and coercivity. The eutectic point for Fe-B binary alloys is around 17% B content [5]. By lowering the B content, the crystallization temperatures of α-Fe and Fe_2_B phases separate, which is required for a proper annealing process. Obtaining an amorphous alloy with an amount of B below 13% is almost impossible due to the low content of metalloid [6]. Nowadays Fe-B binary alloys are widely tested in URA processes. [7]. Parsons et al. in their work obtained Fe_86_B_14_ alloy dedicated for URA and studied the influence of annealing time on the structure and properties of alloys [8]. Similar work was presented by K. Suzuki et al. In this work, Fe-B alloys with various amount of Cu and Nb content with high magnetic saturation up to 1.92 T and low coercivity <8 A/m were presented [9].

Influence of Cu content on structure and magnetic properties in Fe-based alloys were widely studied [10,11,12]. Generally, Cu up to 1.5% plays a role as a grain refiner, providing fine and homogenous structure. Cu nanoclusters act as nucleation sites for the α-Fe crystallites. Additionally, small content of Cu can increase GFA when the main phase is not α-Fe [13]. However in Fe-based alloys, the Cu content reduces GFA due to the positive mixing enthalpy of Fe-Cu system, equal to 13 kJ/mol [14]. Although, there is some data on the effect of Cu content on the Fe-B system, there is no comprehensive study showing the correlation of thermodynamic parameters with crystallization kinetics, structure and magnetic properties as a function of Cu content and annealing temperature. In previous studies, a thermodynamic approach and correlation between thermodynamic calculations and crystallization kinetics, structure and magnetic properties of metallic glass for the Fe-Co-Mo-B-Si system were presented [15,16].

The main aim of this work is to describe the effect of Cu content (0, 0.4, 0.55, 0.7, 1 at.%) on the crystallization kinetics, structure and magnetic properties of Fe-B alloys in order to find correlations between them and thermodynamics parameters. The chemical composition of the alloys has been optimized according to the minimum of the Gibbs free energy value of the amorphous phase formation. Additionally, the annealing process was optimized in accordance with the criterion of the lowest value of core power loss determined for toroidal cores measured at 1T magnetic induction and 50 Hz frequency.

## 2. Materials and Methods

Precursors for amorphous Fe_86-x_Cu_x_B_14_ (x = 0, 0.4, 0.55, 0.7, 1) alloys were prepared from pure chemical elements Fe (3N), Cu (4N) and the binary compound FeB_18_ (2.5N) with using induction furnace in an argon atmosphere (heating at 1400–1450 °C for 15 min, casting at 1200–1260 °C). The amorphous alloys in the form of ribbons with 6–7 mm width were obtained by melt spinning technique with use of 30 m/s Cu wheel speed and casting temperature in the range from 1200 to 1260 °C. To optimize (minimum value) the core power loss (Ps), the wound toroidal cores were isothermally annealed in a vacuum furnace (5 × 10^−3^ mbar) for 20 min at a various temperatures from 260 to 400 °C. The structures of as-spun and annealed materials were verified by X-ray diffraction (XRD) at room temperature using a Rigaku MiniFlex 600 diffractometer (Rigaku, Tokyo, Japan) equipped with copper tube CuKα (λ = 1.5406 Å). The transmission electron microscopy (TEM) images in the bright-field (BF) mode, dark field (DF) mode and selected area diffraction patterns (SADPs) were recorded for the samples annealed at the temperature of optimal Ps value and 370 °C using Tecnai G2 F20 (200 kV) electron microscope (Thermo Fisher Scientific, Waltham, MA, USA). The kinetic of crystallization process was determined based on differential scanning calorimetry (DSC) measurements, made at heating rates from 5 to 30 °C/min using Netzsch DSC 404C Pegasus thermal analyzer (NETZSCH-Gerätebau GmbH, Selb, Germany). Magnetic properties studies at room temperature of toroidal cores (20 mm internal diameter, 30 mm outer diameter, 10–12 g weight) at 50 Hz were performed using Remacomp C-1200 magnetic measurement system (MAGNET-PHYSIK Dr. Steingroever GmbH, Köln, Germany). Coercivity (*Hc*), magnetic saturation (*Bs*), remanence (*Br*) and squareness factor (*Sf*) were obtain from hysteresis loop measured up to saturation state, the *Ps* were calculated from measurement at B = 1 T (P_10/50_). For samples with optimal *P*_10/50_ value, the measurement of complex magnetic permeability at room temperature and in the frequency range f = 10^4^–10^8^ Hz of toroidal cores (8 mm internal diameter, 20 mm outer diameter) was performed using Agilent 4294A impedance analyzer (Agilent, Santa Clara, CA, USA).

## 3. Results

### 3.1. Thermodynamics Calculations

For the better understanding influence of Cu content in Fe_86-x_Cu_x_B_14_ on kinetics, structure and magnetic properties three thermodynamics parameters i.e., configurational entropy (Δ*S^config^*), Gibbs free energy of mixing (Δ*G^mix^*) and Gibbs free energy of amorphous phase formation (Δ*G^amorph^*) were calculated according to Equations (1)–(7) [16]:(1)$$\Delta H_{k}^{\mathrm{mix}}=4\Delta H_{\mathrm{ij}}^{\mathrm{mix}}c_{i}c_{j}$$
(2)$$\Delta H^{mix}=\sum_{k=1}^{N}\Delta H_{k}^{mix}$$
(3)$$\Delta H_{k}^{\mathrm{amorph}}=4\Delta H_{\mathrm{ij}}^{\mathrm{amorph}}c_{i}c_{j}$$
(4)$$\Delta H^{\mathrm{amorph}}=\sum_{k=1}^{N}\Delta H_{k}^{\mathrm{amorph}}$$
(5)$$\Delta S^{conf}=-R\sum_{i=1}^{n}c_{i}lnc_{i}$$
(6)$$G^{mix}=\Delta H^{mix}-T\Delta S^{conf}$$
(7)$$\Delta G^{amorph}=\Delta H^{amorph}-T\Delta S^{conf}$$
where: $\Delta H_{ij}^{mix}$ is the mixing enthalpy between *i*-th and *j*-th chemical elements for equiatomic composition in a binary system, *c_i_* and *c_j_* are the concentrations of *i*-th and *j*-th elements, *k* is the atomic pair number, *N* is the number of different atomic pairs *ij* (*N* = 15 for *n* = 6), *n* is the number of chemical elements in alloy (in this study *n* = 6), $H_{ij}^{amorph}$ is the amorphization enthalpy between *i*-th and *j*-th chemical elements for equiatomic composition in a binary system, *R* is the gas constant and *T* is the average casting temperature of alloy from liquid state.

The results of thermodynamic calculations are presented in Figure 1. The stable amorphous phase can be obtained when the $\Delta S^{conf}$ parameter is as high as possible whereas $G^{mix}$ and $\Delta G^{amorph}\;$ values are negative. It should be noted, that the addition of copper causes increase in the value of $\Delta S^{conf}$. However, the opposite tendency can be observed for $G^{mix}$, which is associated with the positive mixing enthalpy of the Fe-Cu system. From the other side, the parabolic changes in the $\Delta G^{amorph}$ can be observed. The most negative value of $\Delta G^{amorph}$ was observed at 0.55% of Cu. This minimum can be related to the balance between the enthalpy of formation of amorphous phase and the configurational entropy. Although, the Fe_85.45_Cu_0.55_B_14_ composition was found to be optimal for amorphization, the samples prepared for further studies contained from 0 to 1 at.% of Cu. As one can see in Figure 1, the changes in Gibbs free energy of amorphization are small for alloys with Cu content in the range 0.4 to 1 at.% The most visible changes can be noted for alloys with a copper content in the range 0–0.4 at.%.

### 3.2. Study of as-Spun Materials

To determine the structure of as-spun ribbons XRD was performed. The XRD patterns, presented in Figure 2a, confirmed amorphousness of all materials. Calorimetric study was performed to determine the influence of Cu content on crystallization kinetics and to find a correlation between kinetics and the thermodynamic parameters. Two DSC peaks observed in Figure 2b can be related to the formation of the two phases in the alloys: α-Fe (T_x1_) and borides (T_x2_). The exception is the sample with 0.55% of Cu, where two different temperature of borides phase formation can be noticed. The appearance of two peaks can be associated with the smallest $\Delta G^{amorph}$ value. The temperatures of α-Fe and boride phases crystallization and the difference between T_x1_ and T_x2_ (dT_x_ = T_x2_ − T_x1_) determined from DSC curves have been presented in Table 1. The addition of copper generally increases the onset and peak temperatures of crystallization of both phases by approximately 2.5%, however for the alloy with Cu content equal to 0.55% the temperatures remain unchanged. For this material (Fe_85.45_Cu_0.55_B_14_) the characteristic splitting of one main second crystallization peak (borides) into two can be observed, which is probably related to the crystallization of two different boride phases (Fe_2_B and Fe_3_B). Consequently, the boride peak height is much lower for composition with Cu content equal to 0.55%, compared to other studied alloys.

For better understanding crystallization process the crystallization kinetics studies were performed. In order to determine average activation energy of α-Fe phase crystallization process Kissinger model was used (Equation (8)). The results, together with the uncertainty of measurement have been shown in Table 2 and Figure 3a [17].
(8)$$\ln\left(\frac{\beta }{T^{2}}\right)=-\frac{E_{a}}{RT}+C_{1}$$
where: *β* is the heating rate, $E_{a}$ is the activation energy, *R* is the gas constant, *T* is the temperature of maximum of crystallization peak and $C_{1}$ is the constant. The $E_{a}$ is obtained from the slope of linearly fitted $\ln\left(\beta /T^{2}\right)$ vs. 1/T curves.

The activation energy derived from the Kissinger model is an average activation energy. It means that it takes into account both nucleation and growth process. The addition of Cu generally increases values of $E_{a}$. The exception is the composition with 0.55% copper for which the average activation energy is comparable to the composition without copper.

In order to determine changes in activation energy during the crystallization process of α-Fe phase, the Ozawa-Flynn-Wall (OFW) method was applied [17]:(9)$$ln\left(\beta \right)=-\frac{1.052E_{\alpha }}{{RT}_{\alpha }}+C_{2}$$
where: $T_{\alpha }$ is the temperature corresponding to the certain crystallized fraction (α) and *C*_2_ is constant. The volume of the crystallized fraction α is a ratio of the part area before $T_{\alpha }$ and total area of DSC peak.

The results of OFW calculation for the crystallization fraction in the range of 0.1–0.9 have been presented in Figure 3b. Low values of the $E_{a}$ at the start of crystallization is associated with easy and fast nucleation process. With crystallization progress, the $E_{a}$ increases, which can be connected with slower crystallite growth. In a model compound $E_{a}$ increases at the beginning of crystallization and decreases during this process. For the studied ribbons with Cu content equal to 0.4%, 0.55% and 1% $E_{a}$ significantly increases at the end of the process. Only for alloys with 0% and 0.7% of Cu $E_{a}$ values typically decrease at 0.9 crystallization fraction. This behavior can be associated with the onset of boride crystallization in the final stage of α-Fe crystallization. Moreover, the $E_{a}$ for alloy with Cu = 0.55% is a linear function of crystallization fraction in the full studied range. This material have also the smallest values of $E_{a}$ during the whole crystallization process from all studied alloys. This can be related to the ultrafast nucleation of many crystallites of α-Fe, which block the diffusion process of Fe in the amorphous matrix during annealing.

To complete the kinetic studies, the Augis and Bennett formula [18] was used in order to obtain Avrami exponents (n) describing crystallization mechanism:(10)$$n=\frac{2.5}{FWHM}\cdot \frac{T_{x1}^{2}}{\frac{E_{a}}{R}}$$
where: *T_x_*_1_ is the temperature of maximum peak of α-Fe phase crystallization, FWHM is the full width at half maximum of that peak and $E_{a}$ is the activation energy calculated from Kissinger formula. The average n values for the heating rate equal to 20 °C/min are presented in Figure 3c. The value of n changes from 1.54 to 1.7 for analyzed alloys, which indicates the diffusion controlled growth with decreasing nucleation rate mechanism.

### 3.3. Magnetic Properties

The annealing process was carried out in a wide temperature range to determine the optimal temperature, which allows obtaining the best (lowest) value of core power losses. In Figure 4 the core power loss P_10/50_ (a) measurement at the induction of 1T and frequency 50 Hz together with magnetic saturation Bs (b) and coercivity Hc (c) measured at 50 Hz in the function of annealing temperature Ta were presented. The optimal Ta and corresponding magnetic properties have been collected in Table 3.

Sample with 0.4% of Cu has the worst soft magnetic properties among all alloys. Initial addition of Cu increases the P_10/50_ and Hc values. The Ta initially decreases from 290 °C to 280 °C with 0.4% Cu added to the alloy and then with further copper addition, Ta increases to 300 °C for 1% Cu. The best P_10/50_ parameter has alloy with 0.7% of Cu annealed at 290 °C. However the sample with 1% of Cu has nearly the same P_10/50_ but higher Bs = 1.6 T. Even the low annealing temperature Ta provides an improvement in magnetic properties compared to as-spun materials. Alloys annealed at temperatures higher than 310–320 °C have much worse soft magnetic properties, which is caused by the crystallization of Fe_2_B and Fe_3_B phases. The Hc value increases up to 3884 A/m for alloy without Cu and up to 4517 A/m for the sample with 1% of Cu. For annealed samples at Ta above 320 °C the Bs values fluctuate due to a crystal growth and increase, but level of Hc and Ps values are too high from the application point of view. Thus the optimal heat treatment conditions are from the Ta range 280–300 °C for all compositions.

Figure 5 presents remanence Br (a) and squareness factor Sf (b), calculated from the formula: Br/Bs·100% in the function of annealing temperature Ta. It can be noted, that all of the alloys have similar values of Br and Sf, where the alloy with 0.55% Cu has the highest both of these values at the lower Ta. The exception is the functional behavior of Br(Ta) and Sf(Ta), for Cu = 0.4%. At first, both parameters do not increase, but decrease and reach a minimum at 310°C. The low values of these both parameters can provide application possibility, where the more linear B(H) behavior is desired. Low Sf values are associated with the isotropic nature of the sample and the reduced degree of crystallographic alignment, which significantly changes due to the crystallites growth of boride phases. The hysteresis loop for the sample annealed at the optimum Ta has been shown in Figure 6a and it corresponds to previous results.

Magnetic permeability µ’ and magnetic permeability loss µ” dependence in the function of frequency 10^4^–10^8^ Hz for annealed samples at the optimum Ta have been presented in Figure 6b. One can distinguish two permeability behaviors due to the amount of copper. The initial addition of Cu up to an amount of 0.55% causes a decrease in the values of µ’ and µ”. The permeability values for this amount of copper (0.55%) are the lowest (static permeability in the low frequency limit is equal to 750 compared to 1700 for copper-less alloy). Adding Cu above 0.55% causes increase of permeability values. Static permeability for the 0.7% and 1% Cu is equal to 1800 and 2300, respectively. It is worth mentioning, that in a sample with a content of 0.4% Cu the cut-off frequency occurs at a higher frequency than in a sample containing 1% of Cu.

### 3.4. Structure after Annealing

The influence of annealing temperature on crystal structure (visualized by XRD patterns) of Fe_86-x_Cu_x_B_14_ alloys has been presented in Figure 7. There are three scenarios of atomic arrangements in the materials. Initially, as-spun alloys are characterized by anamorphic structure with only short-range ordering. With annealing at lower temperatures, the process of spontaneous relaxation is accelerated. Stresses accumulating in the manufacturing process are subject to relaxation—the materials have a relaxed glass structure. As the annealing temperature increases, the first α-Fe nucleation site appears. Materials with α-Fe nanocrystallites in the amorphous matrix have the best soft magnetic properties. At 340 °C and above, crystallization of boride phases occurs. At this temperature, the materials have a full crystalline structure with close-range ordering. The intensity of the boride phase peaks depends on the annealing temperature. At 340 °C mainly the Fe_2_B phase occurs, while at higher temperatures Fe_3_B phase appears. This is particularly noticeable in samples with the addition of Cu. Samples processed at the optimum Ta are in an amorphous state as relaxed metallic glass.

The BF images and SADPs patterns of the annealed sample at the optimum Ta and at 370 °C recorded by TEM were presented in Figure 8. The TEM studies (a,b,e,f,i,j) confirmed that the alloys annealed at optimal Ta are mainly characterized by an amorphous matrix with a small amount of α-Fe phase nanocrystals. The TEM studies of samples annealed at 370 °C (c,d,g,h,k,l) showed a fully crystalized structure of the mixture of α-Fe and boride phases. It should be noted, that in the sample without Cu content only α-Fe and Fe_3_B phases occur. In both studied samples containing Cu, in addition to previously mentioned Fe_3_B phase, there is an additional Fe_2_B phase. The Fe_2_B crystallites can be distinguished due to stacking faults presence being typical for Fe_2_B phase and was reported by Shahri et al. [19] and Goldfarb et al. [20].These results correlate with XRD studies, where only for Fe_86_B_14_ alloys annealed at 370 °C the XRD pattern has clearly visible peaks from the Fe_2_B phase. The formation of Fe_2_B and Fe_3_B phases seems to depend on Ta, and the crystallization temperature of both phases is slightly different. Additionally, the mean grain size of α-Fe crystallites for Fe_86_B_14_, Fe_85.45_Cu_0.55_B_14_ and Fe_85_Cu_1_B_14_ ribbons heat treated at 370 °C has been manually estimated comparing both bright field and dark field images. Measurements have been performed for around 100 crystallites for each sample. The measured values of α-Fe crystallites were found to be 105.5 ± 35.6 nm, 98.9 ± 33.9 nm and 84.6 ± 25.6 nm for Fe_86_B_14_, Fe_85.45_Cu_0.55_B_14_ and Fe_85_Cu_1_B_14_, respectively. Thus, one can see that with the increase of the amount of Cu, the α-Fe crystallites decrease. Figure 9 shows exemplary bright field and set of dark field images of Fe_85.45_Cu_0.55_B_14_ annealed at 370 °C, taken from a brighter reflections of the hkl rings corresponding to α-Fe phase (marked by red circles in SADP image). To provide the same conditions of acquisition for each sample all DFs were taken from similar areas with respect to a hole in the thin foil. It can be seen that α-Fe crystallites were differentiated and consequently carefully included for the measurement purpose while overlapping crystallites were excluded. It can be stated that observations performed using DF mode were useful to clear distinction of α-Fe crystallites and calculation of their size being simultaneously a supplement of BF images.

### 3.5. Discussion

The effect of Cu content and Gibbs free energy of amorphous phase formation ΔG^amorph^ on the crystallization kinetics and magnetic parameters for annealed samples at optimal Ta has been shown in Figure 10. There is a correlation between Gibbs free energy of amorphous phase formation ΔG^amorph^ and crystallization kinetics (Figure 10a). The most negative value of ΔG^amorph^ corresponds to the lowest value of crystallization temperature and to the average E_a_ of α-Fe phase formation. The average activation energy obtained by the Kissinger method for the binary compound Fe_86_B_14_ equal to 201.8 kJ/mol is lower compared to the literature data. In their work, Parsons et al. have obtained activation energy an amorphous Fe_86_B_14_ ribbon equal to 2.6 eV ≈ 250.9 kJ/mol [8].

Magnetic measurements (Figure 10b–d) showed that a small addition of Cu causes a deterioration of soft magnetic properties (P_10/50_ ↑, Hc↑, Bs↓, µ’↓), but shifts the cut-off frequency towards higher frequencies. On the other hand, alloys with Cu content above 0.55% have better soft magnetic properties (P_10/50_↓, Hc↓, Bs↑, µ’↑) than the sample without Cu. Materials annealed at optimal temperature had P_10/50_ equal to 0.13–0.25 W/kg, Bs equal to 1.47–1.6 T and Hc equal to 9.71–13.1 A/m. Similar Fe-based alloys were investigated by Zang et al. [6]. In a 5 min annealing process with a heating rate 1.7 °C/s the authors have obtained Fe_85.5_B_13_Cu_1.5_ and Fe_87_B_13_ (at the optimum Ta) Hc values equal to 14.5 A/m and above 25 A/m, respectively. By performing URA process (3 s of annealing at 485 °C with heating rate 150 °C /s) the magnetic properties have been significantly improved and were equal to 3.0 A/m and 1.86 T for Fe_85.5_B_13_Cu_1.5_ and 6.7 A/m and 1.92 T for Fe_87_B_13_ respectively. Similar results were described by the Suzuki et al., where URA processed Fe_86_B_14_ and Fe_86_B_13_Cu_1_ ribbons had parameters equal to 3.5–7.5 A/m, 1.89 T and 0.19–0.38 W/kg at P_15/50_, respectively [7]. The magnetic properties of Fe-B-Cu alloys after the URA process looks promising and further research into our materials in this field should be considered. The correlations between magnetic properties and thermodynamic parameters have been found. The static magnetic permeability clearly corresponds to the Gibbs free energy of amorphous phase formation. For the Cu content equal to 0.55%, the $\Delta G^{amorph}$ has lowest value −20.03 kJ/mol. For this composition static permeability is equal to 750. The highest amorphization tendency is probably related to the formation of smaller magnetic domains with smaller resultant magnetic moment.

## 4. Conclusions

In this paper the influence of Cu content on Fe_86-x_Cu_x_B_14_ alloys was determined. Moreover, our results provide evidence of a correlation between thermodynamic parameters, especially the Gibbs free energy of amorphous phase formation, and the kinetic, structural and magnetic properties of alloys. Thermodynamic calculations have shown that a material with Cu = 0.55% has the lowest energy of amorphous phase formation. This sample is characterized by the lowest onset and peak *T_x_*_1_ crystallization temperatures equal to 380.9 °C and 405.1 °C, respectively. Moreover sample with this composition has the lowest average activation energy calculated by Kissinger formula (202.56 ± 1.28 kJ/mol).The activation energy calculated on the basis of the OFW formula is almost lowest for each crystallization fraction. The small addition of Cu deteriorates the soft magnetic properties of the alloy, however above 0.55% Cu magnetic properties of the samples were improved compared to the sample without Cu. The optimization of annealing temperature showed that the material with 0.7% Cu content annealed at 290 °C has the best *P*_10/50_ = 0.13 W/kg with *Bs* = 1.53 T and *Hc* = 9.71 A/m. Complex permeability measurement of samples annealed at optimal *P*_10/50_ value showed, that the sample with Cu = 0.55% is in the minimum of *µ’* and *µ”* values. XRD and TEM studies of samples annealed at optimal *P*_10/50_ confirmed its amorphousness with the presence of a small amount of α-Fe nanocrystals.

## Figures and Tables

**Figure 1 materials-13-01451-f001:**
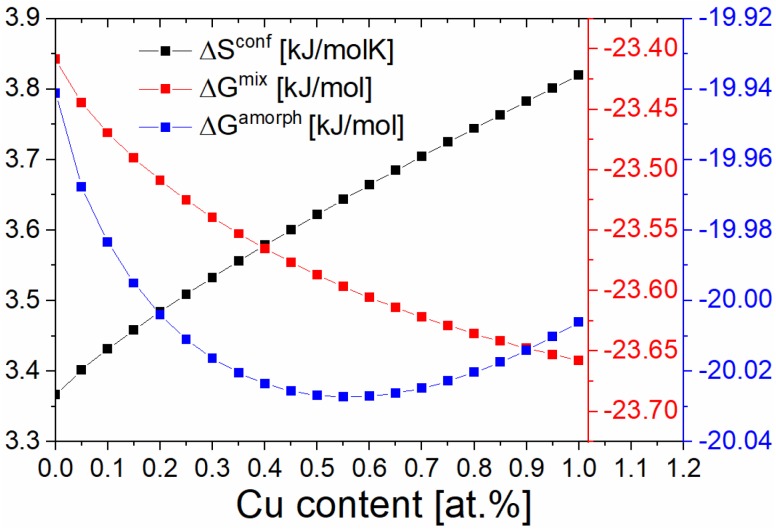
Dependencies of thermodynamic parameters as a function of Cu content.

**Figure 2 materials-13-01451-f002:**
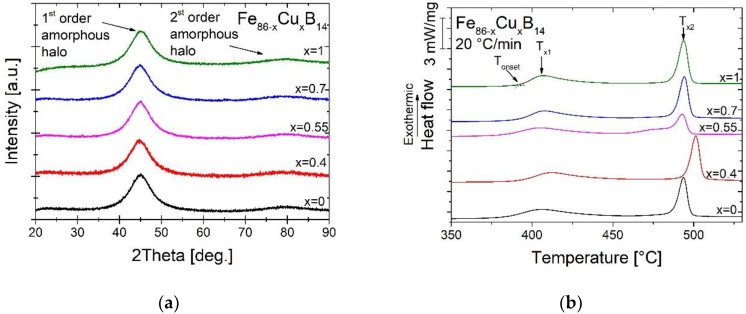
XRD patterns (**a**) and differential scanning calorimetry (DSC) signals (**b**) for as-spun metallic glasses.

**Figure 3 materials-13-01451-f003:**
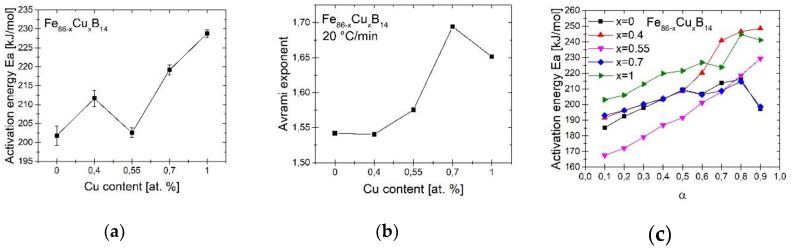
Influence of Cu content on: average activation energy Ea (**a**), fraction dependent activation energy of α-Fe phase crystallization (**b**) and Avrami exponent (**c**).

**Figure 4 materials-13-01451-f004:**
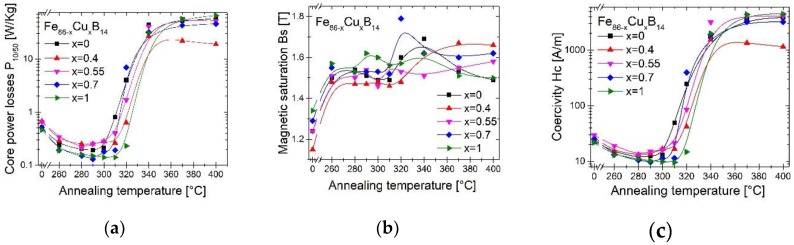
Core power losses P_10/50_ (**a**), magnetic saturation Bs (**b**) and coercivity Hc (**c**) dependence on annealing temperature Ta.

**Figure 5 materials-13-01451-f005:**
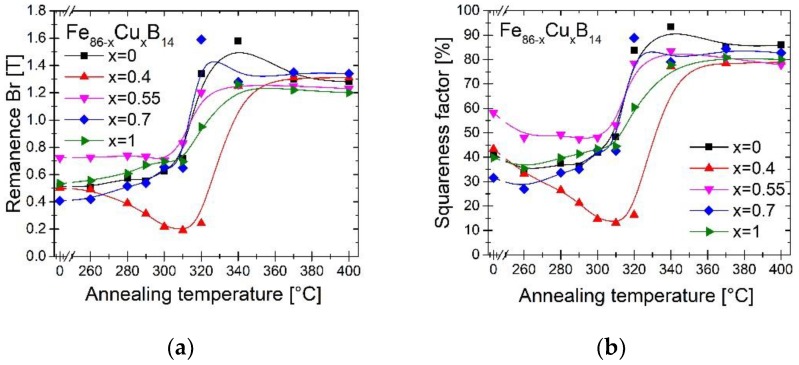
Remanence Br (**a**) and squareness factor Sf (**b**) dependence on annealing temperature Ta.

**Figure 6 materials-13-01451-f006:**
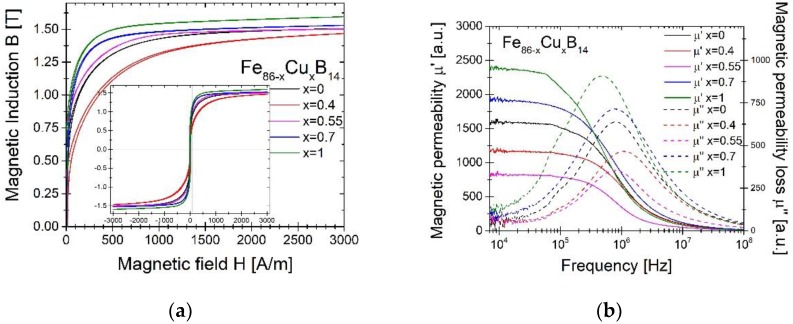
Hysteresis loops: (**a**) real µ’ and imaginary µ” part of permeability; (**b**) dependence in the function of frequency 10^4^–10^8^ Hz for samples annealed at the optimum Ta.

**Figure 7 materials-13-01451-f007:**
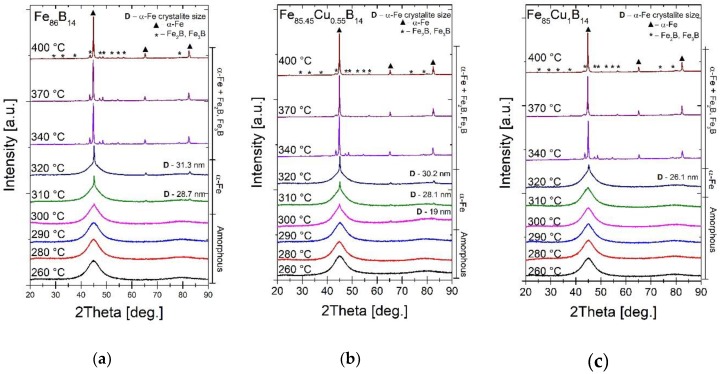
XRD patterns for annealed alloys: Fe_86_B_14_ (**a**), Fe_85.45_Cu_0.55_B_14_ (**b**), Fe_85_Cu_1_B_14_ (**c**).

**Figure 8 materials-13-01451-f008:**
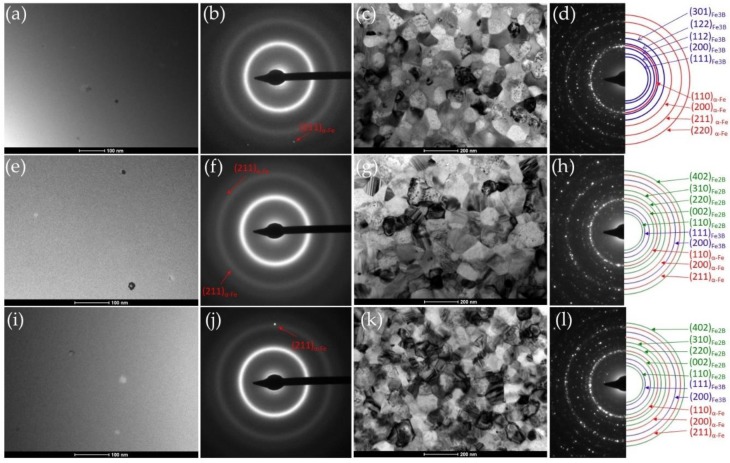
TEM images of annealed at temperatures at optimal P_10/50_ values and full-crystalized samples: Fe_86_B_14_ at 290 °C bright-field (BF) (**a**), selected area diffraction patterns (SADP) (**b**) and at 370 °C BF (**c**), SADP (**d**); Fe_85.45_Cu_0.55_B_14_ at 280 °C BF (**e**), SADP (**f**) and at 370 °C BF (**g**), SADP (**h**); Fe_85_Cu_1_B_14_ at 300 °C BF (**i**), SADP (**j**) and at 370 °C BF (**k**), SADP (**l**).

**Figure 9 materials-13-01451-f009:**
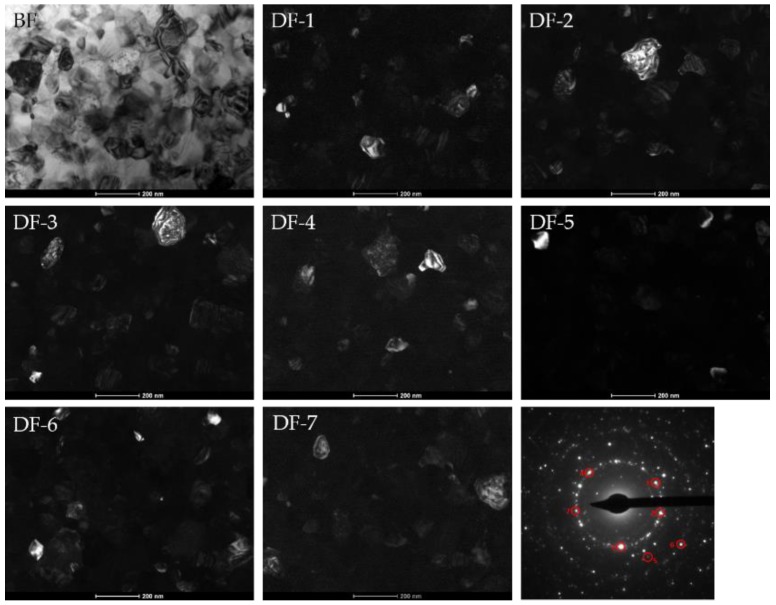
Bright field image, set of dark field images and selected area diffraction patterns of Fe_85.45_Cu_0.55_B_14_ annealed at 370 °C.

**Figure 10 materials-13-01451-f010:**
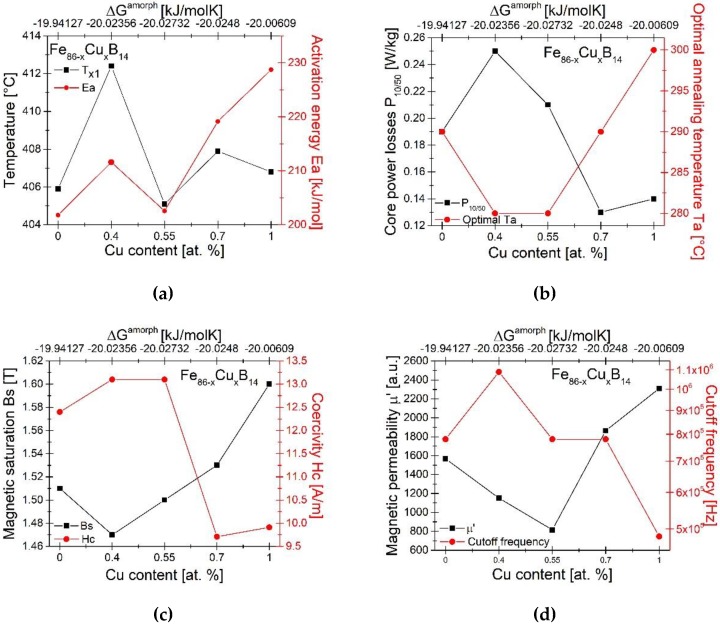
Influence of Cu content and Gibbs free energy of amorphous phase formation ΔG^amorph^ on: characteristic crystallization temperatures at heating rate 20 °C/min and average activation energy Ea (**a**), optimal annealing temperature Ta and magnetic properties at optimal Ta: core power losses P_10/50_ (**b**), magnetic saturation Bs and coercivity Hc (**c**), magnetic permeability µ’ and cutoff frequency (**d**).

**Table 1 materials-13-01451-t001:** The characteristic temperatures of crystallization at heating rate 20 °C/min.

Alloy	T_onset_ (°C)	T_x1_ (°C)	T_x2_ (°C)	dT_x_ (°C)
Fe_86_B_14_	382.4	405.9	493.8	87.9
Fe_85.6_Cu_0.4_B_14_	391.6	412.4	501.3	88.9
Fe_85.45_Cu_0.55_B_14_	380.9	405.1	492.9	87.8
Fe_85.3_Cu_0.7_B_14_	390.4	407.9	494.1	86.2
Fe_85_Cu_1_B_14_	391.5	406.8	493.9	87.1

**Table 2 materials-13-01451-t002:** The activation energy $E_{a}$ calculated from the T_x1_ temperatures with errors.

Alloy	$E_{a\;}(\mathrm{kJ}/\mathrm{mol})$	Error (kJ/mol)
Fe_86_B_14_	201.8	2.57
Fe_85.6_Cu_0.4_B_14_	211.6	2.12
Fe_85.45_Cu_0.55_B_14_	202.56	1.28
Fe_85.3_Cu_0.7_B_14_	219.15	1.3
Fe_85_Cu_1_B_14_	228.74	1.01

**Table 3 materials-13-01451-t003:** Magnetic properties of alloys for optimal Ta.

Alloy	Ta (°C)	P_10/50_ (W/kg)	Bs (T)	Hc (A/m)
Fe_86_B_14_	290	0.19	1.51	12.4
Fe_85.6_Cu_0.4_B_14_	280	0.25	1.47	13.1
Fe_85.45_Cu_0.55_B_14_	280	0.21	1.5	13.1
Fe_85.3_Cu_0.7_B_14_	290	0.13	1.53	9.71
Fe_85_Cu_1_B_14_	300	0.14	1.6	9.91

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
