# Peer review of "Influence of Cu Content on Structure and Magnetic Properties in Fe86-xCuxB14 Alloys"

_materials, 2020, doi:10.3390/ma13061451_

Round 1
Reviewer 1 Report
The paper presents the influence of Cu content on thermodynamic parameters, crystallization kinetics, structure and magnetic properties of Fe-Cu-B alloys. Some results are interesting. I have some suggestions for reconsideration. 1. As shown in Tal.3, for the sample of 1.0% Cu, annealing at 300°C provided the best magnetic properties. Considering the different copper content and annealing temperature, in order to reveal the structural factors in iron-based amorphous alloys that have a crucial effect on magnetic properties, the author should give a more detailed analysis of the evolution of microstructure with the different copper content and annealing temperature. 2. Figure 9 shows exemplary bright field and set of dark field images of Fe85.45Cu0.55B14 annealed at 370°C. The author should explain briefly. 3. Some figures, such as Fig. 3-6, are not of high quality. The author needs to improve.Author Response
Please see the attachment.

Reviewer 2 Report
This is a very useful report with detailed information and interesting results. I strongly recommend the acceptance after minor corrections.
Please consider the following corrections:
In the abstract this sentence should be corrected ". were described"
The same error is repeated (wrong punctuation) in other parts of the manuscript. Therefore Englisch corrections are required.
The advantages of the method used for sample preparation should be explained. References should be included.
For the graphic 2 with XRD, please include peak index.
Font size of text in the figures is too small.
Reviewer 3 Report
This manuscript presents a thorough analysis of the effects of Cu doping and thermal post-processing on crystallization kinetics, structure, and magnetic properties of Fe-B amorphous alloys. The authors study this system theoretically and experimentally, applying various approaches from thermodynamic calculations to measurements of functional magnetic characteristics. The performed study is not original in itself but very useful to the metallic glass field. The paper is well organized and gives a good overview of the issues considered. Nevertheless, some points of the present version of the manuscript need clarifications before publishing:
- As follows from the results, the quenched alloys demonstrate a completely amorphous structure. In this case, the glass transition (Tg) should be easily detected with DSC. Did the authors detect Tg temperatures from the DSC traces?
- The parameter dTx listed in Table 1 should be explained.
- Glass-forming ability (or GFA) and thermal stability of the alloys are analyzed with only two parameters, Tx1 and Tx2. As is well known, there is a large number of very popular thermal criteria (Inoue’s ones for instance) for characterizing the amorphous phase. I think that at least one of the typical parameters like the reduced glass transition Trg = Tg/Tm needs to be estimated to describe the glassy state more correctly.
- There are some typos in this manuscript version. In particular, please pay your attention to these phrases:
…“properties. were described”
…“10-12g wage”.
Please try to revise the manuscript carefully to polish your submission and to avoid any mistakes.
